# Effects of manufacturing modality, primer, and adhesive polymerization on the shear bond strength of customized lingual brackets to glazed zirconia: An *in vitro* study

Viet Anh Nguyen[1]*, Ngo The Minh Pham[1], Minh Ngoc Tran[1], Thi Bich Ngoc Ha[1], Thi Quynh Trang Vuong[2]

**1** Faculty of Dentistry, Phenikaa University, Hanoi, Vietnam, **2** Private Practice, Viet Anh Orthodontic Clinic, Hanoi, Vietnam

* anh.nguyenviet1@phenikaa-uni.edu.vn

## Abstract

### Introduction

Bonding fixed appliances to zirconia restorations is challenging, yet adult orthodontics increasingly involves ceramic crowns and patient-driven esthetic choices such as lingual appliances. Customized lingual brackets may improve fit and reduce adhesive thickness, but evidence on their bonding to zirconia is limited.

### Materials and methods

This *in vitro* study evaluated the shear bond strength of customized lingual brackets bonded to glazed zirconia after airborne-particle abrasion. Bracket manufacturing was either three-dimensionally (3D) printed cobalt-chromium or cast nickel-chromium. Primers were a universal adhesive (Single Bond Universal, 3M) or a primer containing 10-methacryloyloxydecyl dihydrogen phosphate Z-Prime Plus (Bisco), and adhesives were a light-cure orthodontic composite or a dual-cure resin cement. One hundred twenty-eight specimens (n = 16 per group) were tested. Shear bond strength was analyzed with three-way ANOVA, followed by post-hoc Tukey tests. Adhesive Remnant Index (ARI) scores were evaluated with ordinal regression. Significance was set at α = 0.05.

### Results

Manufacturing modality significantly affected bond strength, with additively manufactured cobalt-chromium exceeding cast nickel-chromium (P = 0.049). The primer category and polymerization mode showed no significant main effects (P > 0.20) and no significant interactions. Group means clustered 9–10 MPa, and all combinations met the clinically accepted threshold. Additively manufactured brackets exhibited lower

**Data availability statement:** All relevant data are within the paper and its Supporting Information files.

**Funding:** The author(s) received no specific funding for this work.

**Competing interests:** The authors have declared that no competing interests exist.

ARI scores than cast brackets (P < 0.001), indicating failures closer to the zirconia–adhesive interface. The fabrication×primer term was significant for ARI (P = 0.017).

## Conclusions

On glazed, sandblasted zirconia, shear bond strength of customized lingual brackets showed a borderline main effect of fabrication method, whereas primer type and adhesive polymerization mode were not statistically significant. Failures were predominantly located at or near the zirconia–adhesive interface. Within this *in vitro* model, base manufacturing may warrant attention, whereas primer and curing mode may be selected for handling and workflow considerations, with clinical relevance yet to be established.

## Introduction

Adult demand for orthodontic treatment has increased in recent years, and esthetics is a major driver of appliance choice [1]. Many adults prefer esthetic options such as lingual appliances or clear aligners [2,3]. Adults also present more frequently with compromised dentitions and existing prosthetic crowns than adolescents. Among ceramic materials, monolithic zirconia is widely used for posterior restorations because of its high flexural strength and the absence of veneering porcelain chipping [4,5].

However, bonding to ceramics requires protocols that differ from bonding to enamel. For glass ceramics, hydrofluoric acid etching followed by coupling agents can provide adequate bond strength [6]. In contrast, zirconia is chemically inert relative to glass ceramics and cannot be predictably etched with hydrofluoric acid. Airborne particle abrasion is therefore recommended to promote micromechanical retention [7]. The functional monomer 10-methacryloyloxydecyl dihydrogen phosphate (MDP) was first introduced as a metal primer and has since been demonstrated to form chemical bonds with zirconia [8]. Digital dentistry enables the fabrication of customized appliances with bracket bases adapted to the lingual tooth surface, which can enhance placement accuracy, minimize adhesive thickness, and potentially increase bond strength [9].

A gap remains in the literature because most bonding studies on ceramic substrates have focused on stock metal or ceramic brackets or on aligner attachments rather than on fully customized bracket systems [7,10–14]. For customized lingual brackets, available *in vitro* work has largely evaluated bonding to enamel [15,16]. To our knowledge, there are no *in vitro* studies that compare additively manufactured brackets with cast counterparts on zirconia while simultaneously testing primer type and adhesive polymerization mode. Despite the growing adoption of additive manufacturing for orthodontic appliances, no study has clarified whether the distinct surface topography and alloy microstructure of three-dimensionally (3D) printed brackets influence bonding behavior on zirconia compared with traditional cast counterparts. Establishing these differences is clinically relevant because mismatched surface

characteristics can alter micromechanical retention and chemical coupling, ultimately affecting bracket reliability on zirconia restorations.

Accordingly, the objectives of this study were to assess the effects of manufacturing modality, primer category, and adhesive polymerization on shear bond strength and failure modes to glazed zirconia, and to test the null hypotheses that none of these factors nor their interactions would affect the measured outcomes. A priori, it was expected that 3D-printed Co–Cr would yield equal or higher bond strength than cast Ni–Cr owing to rougher post-abrasion bases; that an MDP-containing primer would perform equal to or better than a universal adhesive on zirconia via phosphate–oxide coupling; and that dual-cure would perform equal to or better than light-cure on zirconia because of light attenuation in the substrate.

## Materials and methods

### Study design

This *in vitro* study was designed and reported in accordance with the modified CONSORT checklist for laboratory investigations. No human participants, patient data, or animal tissues were involved; therefore, institutional ethical approval was not required. An a priori sample-size calculation was performed with G*Power (version 3.1; Heinrich Heine University Düsseldorf, Düsseldorf, Germany) using F tests (ANOVA: fixed effects, special, main effects and interactions) for a three-way ANOVA ($2 \times 2 \times 2$ between-subjects) with eight groups and numerator degrees of freedom of 1 for each main effect and interaction. The calculation assumed a medium effect size of 0.25 (the smallest effect of interest, based on prior *in vitro* bonding reports), an alpha level of 0.05, and a statistical power of 0.80 [17,18]. The required total sample size was 128, with 16 specimens per group and equal allocation across the eight experimental cells.

### Sample preparation

A premolar crown mesh from the Autolign library (Diorco, Gyeonggi-do, Korea) was selected as the bonding substrate because lingual brackets show higher debonding in posterior teeth, making premolars a clinically relevant site for bond-strength testing. [19] One hundred twenty-eight monolithic zirconia teeth were milled by computer-aided manufacturing from 3-mol% yttria-stabilized tetragonal zirconia polycrystalline (3Y-TZP) discs (Superfect Zir, Aidite, Hebei, China) using a 5-axis AMD-500E milling machine (Aidite), sintered in a CSF-200 furnace (Aidite), and surface-glazed with BIOMIC low-fusing porcelain (Aidite). All firings followed the manufacturer's instructions.

The premolar STL was also imported into Meshmixer (Autodesk, San Rafael, CA, USA) to design a customized lingual bracket with a cement space of 0.03 mm and overall bracket thickness of 0.30 mm. The bracket base area, measured within Meshmixer, was 19.964 mm². Sixty-four brackets were metal 3D-printed in cobalt-chromium (Co-Cr) alloy using an NCL-M150 selective laser-melting printer (Chamlion, Nanjing, China). The remaining 64 brackets were cast in nickel-chromium (Ni-Cr) alloy from 3D-printed wax patterns produced on a Photon D2 digital light processing printer (Anycubic, Shenzhen, China) with Dental Castable resin (Anycubic). Casting followed a conventional lost-wax workflow.

After fabrication, all supports were removed, and brackets were polished in an electric rotary tumbler for 72 hours (KT-6808; Tasanol, Shenzhen, China). Then, both the glazed zirconia surfaces and the bracket bases were subjected to airborne-particle abrasion using 110-µm aluminum oxide for 20 sec at 0.3 MPa, delivered perpendicular to the surface (90°) from a 10-mm stand-off distance. Under these parameters, the procedure was intended to deglaze and expose the zirconia substrate, avoiding bonding to the glaze, which adheres weakly to zirconia and is prone to debonding [5,20]. Surface roughness (Ra) was then measured with a digital microscope (VHX-7000, Keyence Corporation, Osaka, Japan). Surface roughness of zirconia was recorded as a process-control measure to verify preparation consistency and to contextualize failure modes. Finally, specimens were ultrasonically cleaned in 90% ethanol for 20 min and dried with oil-free compressed air.

## Bonding procedures

After sandblasting and ultrasonic cleaning, specimens were assigned to eight groups defined by bracket fabrication methods (3D-printing versus casting), primer (universal primer versus MDP), and adhesive polymerization (light-cure bracket adhesive versus dual-cure resin cement).

Either a universal primer (Single Bond Universal, 3M Unitek, Monrovia, CA, USA) or an MDP primer (Z-Prime Plus, BISCO, Schaumburg, IL, USA) was applied uniformly to both the bracket base and the glazed zirconia surface, actively agitated for 10 sec, and then air-thinned gently for 5 sec. The primers were not light-cured prior to seating to avoid the formation of a pre-polymerized film that could increase thickness and impair bracket fit [21]. For qualitative surface characterization, an additional six zirconia specimens and six brackets per bracket type were examined by scanning electron microscopy (SEM) (Quanta 450 FEG, Thermo Fisher Scientific, Hillsboro, OR, USA) at 20 kV, including two left unprimed, two treated with Single Bond Universal, and two treated with Z-Prime Plus.

Either a light-cure orthodontic bracket adhesive (Transbond XT, 3M Unitek, Monrovia, CA, USA), applied directly to the bracket base, or a dual-cure resin cement (Overcem, Overfibers, Verona, Italy), mixed per the manufacturer's instructions and then applied, was used (Table 1). In both cases, the bracket was seated on the zirconia surface using a 3D-printed transfer jig under a constant 10-N load, excess was removed with an explorer. Light activation was performed with a Led.F (Woodpecker, Guilin, China) curing unit (tip diameter 8 mm); irradiance at the tip was measured as 1200 mW/cm² with an accompanying radiometer before each specimen. Each margin (mesial, distal, occlusal, and gingival) was light-cured for 20 sec.

All bonded specimens underwent thermocycling to simulate thermal fatigue, with 2000 cycles between 5 °C and 55 °C using a dual-bath thermocycling unit (YTST-021, Yuanyao, Guangdong, China). Each cycle comprised a 25-sec dwell in each bath with a 10-sec transfer time between baths. Based on commonly used laboratory conversions, 2,000 cycles approximate 2.4 months of clinical service [22]. This regimen was selected to represent early intraoral aging, the period in which bracket debonding is most likely to occur clinically [23].

**Table 1. Materials, manufacturers, and full chemical compositions used in this study.**

| Material | Manufacturer | Composition |
|---|---|---|
| Single Bond Universal | 3M | 2-hydroxyethyl methacrylate, bisphenol A diglycidyl ether dimethacrylate, 10-methacryloyloxydecyl dihydrogen phosphate, ethanol, water, 3-(trimethoxysilyl)propyl methacrylate, copolymer of acrylic and itaconic acid (Vitrebond copolymer), camphorquinone, dimethylaminobenzoate, (dimethylamino)ethyl methacrylate. |
| Z-Prime Plus | Bisco | Ethanol, 2-hydroxyethyl methacrylate, bisphenol A diglycidyl methacrylate, 10-methacryloyloxydecyl dihydrogen phosphate, triethylamine. |
| Transbond XT | 3M | Silane-treated quartz, bisphenol A dimethacrylate resins (bisphenol A diglycidyl dimethacrylate and related species), 3-(trimethoxysilyl)propyl methacrylate, diphenyliodonium hexafluorophosphate. |
| Overcem | Overfibers | Urethane dimethacrylate, triethylene glycol dimethacrylate, 2-hydroxyethyl methacrylate, dibenzoyl peroxide, titanium dioxide, bismuth oxychloride, ytterbium fluoride, nanosilica, silanized and ionomeric glass fillers, 4-methacryloxyethyl trimellitic anhydride, 10-methacryloyl-oxydecyl dihydrogen phosphate, bisphenol-A glycidyl dimethacrylate, ultraviolet stabilizer, co-initiators and activators, pigments. |
| Superfect Zir | Aidite | Zirconium dioxide stabilized with 3 mol% yttrium oxide (tetragonal phase), hafnium oxide, aluminum oxide. |
| BIOMIC glaze | Aidite | Silica glass-ceramic, potassium oxide, sodium oxide, calcium oxide, aluminum oxide, opacifiers (titanium dioxide, zirconium dioxide), inorganic pigments, organic binder. |

## Shear bond strength testing and failure mode evaluation

Each bonded zirconia specimen was embedded in autopolymerizing acrylic within a cylindrical mold, leaving the clinical crown exposed and oriented so that the load would be applied parallel to the bracket-zirconia interface. The specimens were clamped in a universal testing machine (HP-1 kN; Handpi Instruments, Shenzhen, China) and loaded with a chisel-edge blade positioned at the bracket base-tooth interface (Fig 1). Load was applied at a crosshead speed of 1.0 mm/min until failure, and the maximum load was recorded in Newtons (N). Shear bond strength was calculated in megapascals (MPa) as the peak load divided by the measured bracket-base area. No support plate was present at the bracket base, and no finishing was performed on the base surfaces; only standardized airborne-particle abrasion was applied. Consequently, the bracket base area was not altered by the fabrication method. Testing followed ISO 29022:2013 notched-edge shear principles, adapted to bracket–zirconia assemblies [24].

Failure mode was evaluated under 5×magnification using the Adhesive Remnant Index (ARI), with scores ranging from 0 to 3 [13,14,25]. Score 0 denotes no adhesive remaining on the zirconia surface, indicating failure at the resin-zirconia interface. Score 1 denotes less than 50% adhesive remaining on zirconia, indicating a predominantly resin-zirconia failure with mixed characteristics. Score 2 denotes more than 50% adhesive remaining on zirconia, indicating a predominantly bracket-adhesive failure with mixed characteristics. Score 3 denotes all adhesive remaining on zirconia, indicating failure at the bracket-adhesive interface.

## Statistical analysis

The primary endpoint was shear bond strength, summarized as mean ± standard deviation. Assumptions for parametric modeling were checked a priori: normality by Shapiro–Wilk tests; variance homogeneity by Levene's test; and outliers by studentized residuals. A three-way factorial ANOVA (2 × 2 × 2) was fitted with bracket fabrication method, primer, and luting material as fixed factors, including all main effects, all two-way interactions, and the three-way interaction. Multiplicity-controlled pairwise comparisons used Tukey's Honestly Significant Difference (HSD) on model-based estimated marginal means for the simple effects of fabrication×primer, evaluated separately within each polymerization condition. In addition, four simple contrasts compared light-cure vs dual-cure within each fabrication×primer cell using t-tests

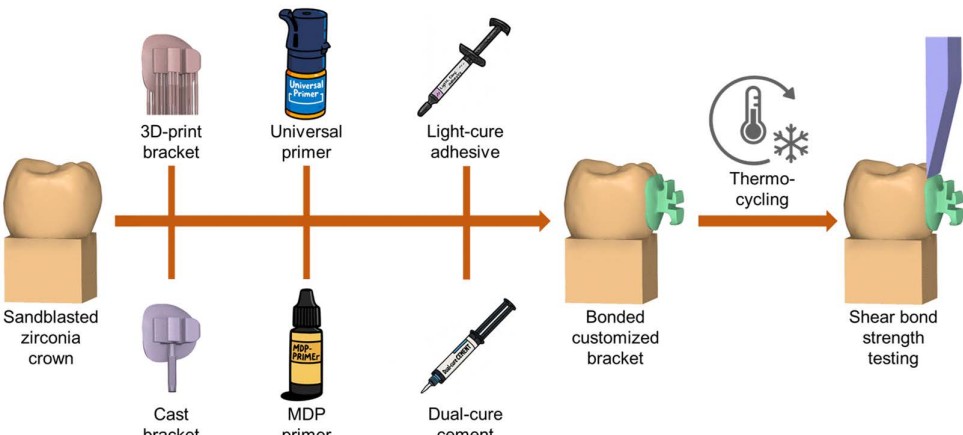

**Fig 1. Study workflow outlining specimen preparation.** Sandblasted zirconia crowns were bonded with customized lingual bracket bases manufactured either by 3D-printing or by casting. Two primers were evaluated: a 10-methacryloyloxydecyl dihydrogen phosphate (MDP) primer and a universal primer. Two polymerization modes were used: a light-cure orthodontic composite and a dual-cure resin cement. After bonding, specimens underwent thermocycling and were tested for shear bond strength.

with Holm adjustment. All tests were two-sided with α = 0.05. Failure mode was analyzed with a proportional-odds ordinal regression including all main effects and interactions up to the three-way term. Two calibrated examiners, blinded to group allocation, independently scored ARI, and inter-rater reliability was quantified using Cohen's κ. Analyses were conducted in Python (version 3.11; Python Software Foundation, Wilmington, DE, USA) using statsmodels (version 0.14), and plots were generated with Matplotlib (version 3.8).

## Results

After sandblasting, the mean [standard deviation (SD)] surface roughness (Ra) was 0.81 [0.15] µm for zirconia, 6.26 [1.33] µm for 3D-printed Co-Cr, and 2.12 [0.43] µm for cast Ni-Cr. A two-sample t-test confirmed higher roughness in Co-Cr (P < 0.001). On scanning electron microscopy, sandblasted surfaces in all three materials exhibited irregular peaks and valleys consistent with micro-texturing. For zirconia surfaces, no continuous glassy glaze layer was observed, consistent with exposure of the underlying zirconia. After application of the MDP primer, selected depressions appeared electron-lucent (dark), consistent with primer pooling within surface valleys (Fig 2). With the universal primer, an almost continuous electron-lucent film covered both metal substrates, whereas on zirconia, the film was nearly confluent with small bright islands of exposed zirconia still visible.

Regarding shear bond strength, the three-way ANOVA (Table 2) revealed a marginal main effect of fabrication method (F = 3.95, P = 0.049), with higher values observed for 3D-printed brackets compared with cast brackets. The main effects of primer (F = 1.39, P = 0.241) and adhesive polymerization (F = 1.65, P = 0.201) were not significant. None of the interaction effects reached significance, including fabrication×primer (F = 0.39, P = 0.536), fabrication×adhesive polymerization (F = 0.94, P = 0.334), primer×adhesive polymerization (F = 0.96, P = 0.330), and fabrication×primer×adhesive polymerization (F = 0.02, P = 0.876).

Across universal and MDP primers, shear bond strength clustered around 9 to slightly above 10 MPa for most combinations, except cast brackets with light-cure adhesive, which were slightly above 7 MPa (Fig 3). Tukey HSD within the light-cure adhesive column indicated higher bond strength for 3D-printed bases than for cast bases for both primers (Table 3). Within the dual-cure adhesive column, there were no between-row differences. Row-wise comparisons of adhesive type were significant only for the cast-universal primer combination, where dual-cure exceeded light-cure (P = 0.003, Holm-adjusted). No within-row differences were detected for the other combinations (P > 0.34).

Inter-rater agreement for ARI scoring was perfect between the two examiners (Cohen's κ = 1.00). In the proportional-odds model with a logit link, fabrication significantly affected ARI, with 3D-printed specimens showing lower ARI levels than cast (OR, 0.06; 95% CI, 0.01–0.25; P < 0.001). A significant fabrication×primer interaction was detected (OR 10.62; 95% CI 1.53–73.61; P = 0.017), indicating that the reduction in ARI with 3D-printed brackets was most pronounced with the MDP primer and attenuated with a universal primer. The main effects of primer and adhesive polymerization were not significant (P = 0.669 and P = 0.931, respectively), and no other interactions reached significance (all P > 0.22). Regarding distribution, ARI 0–1 constituted the majority in 6 of 8 groups (56.2%−93.8%), indicating failure patterns with more adhesive remaining on the bracket (Fig 4). The exception was the universal-light combination, which yielded ARI 0–1 totals of 50.0% for 3D-printed and 25.0% for cast (Fig 5).

## Discussions

This *in vitro* study evaluated whether bracket fabrication method, primer type, and adhesive polymerization affect shear bond strength to glazed zirconia and failure mode. The data partially refute the null hypotheses, showing that the fabrication method significantly increased shear bond strength for 3D-printed brackets over cast brackets, whereas primer type and adhesive polymerization did not exert significant main effects, and no interactions were detected. Failure mode likewise depended on fabrication, with 3D-printed groups tending toward lower ARI scores than cast, an effect accentuated with MDP primer, while other effects were nonsignificant. Complementary surface roughness and SEM findings provide a

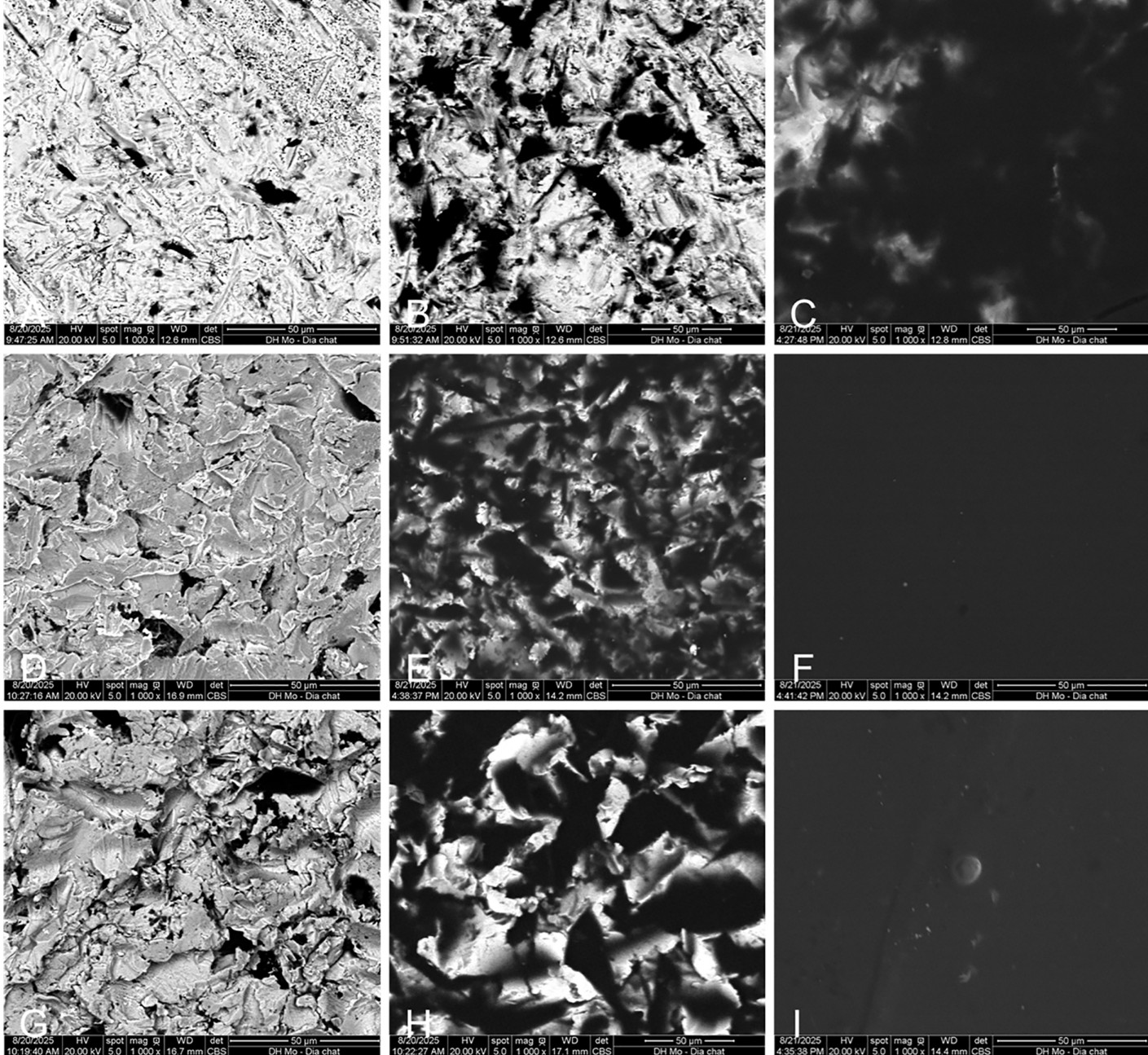

**Fig 2. Representative scanning electron micrographs (1000×). (A–C)** Zirconia surfaces after sandblasting, with 10-methacryloyloxydecyl dihydrogen phosphate (MDP) primer, and universal primer. **(D–F)** 3D-printed bracket bases after sandblasting, with MDP primer, and universal primer. (G–I) Cast bracket bases after sandblasting, with MDP primer, and universal primer.

plausible mechanistic basis. Collectively, these results indicate that manufacturing modality, rather than primer chemistry or curing mode, is the primary driver of bonding performance for customized lingual brackets on glazed zirconia.

In this study, Co-Cr was used for 3D printing because Co-Cr is the workhorse alloy for metal additive manufacturing in dentistry, with high strength and corrosion resistance, and documented accuracy in printed intraoral appliances and

**Table 2. Results of three-way ANOVA on shear bond strength according to fabrication method, primer, and adhesive polymerization.**

| Effect | df | Mean square | F | P value |
|---|---|---|---|---|
| Fabrication | 1 | 30.02 | 3.95 | 0.049 |
| Primer | 1 | 10.58 | 1.39 | 0.241 |
| Adhesive | 1 | 12.55 | 1.65 | 0.201 |
| Fabrication×primer | 1 | 2.94 | 0.39 | 0.536 |
| Fabrication×Adhesive | 1 | 7.15 | 0.94 | 0.334 |
| Primer×adhesive | 1 | 7.28 | 0.96 | 0.330 |
| Fabrication×primer×adhesive | 1 | 0.19 | 0.02 | 0.876 |

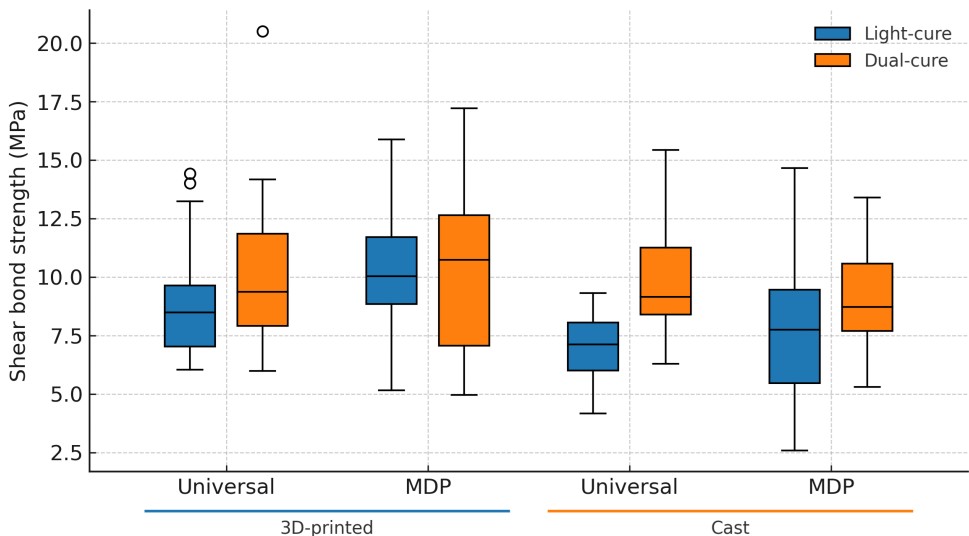

**Fig 3. Box plots of shear bond strength by bracket fabrication modality, primer category, and adhesive polymerization mode.**

**Table 3. Shear bond strength by fabrication method and primer, stratified by adhesive polymerization type.**

| Fabrication method | Primer | Light-cure (MPa) | | Dual-cure (MPa) | |
|---|---|---|---|---|---|
| | | Mean±SD | 95% CI | Mean±SD | 95% CI |
| 3D-printed | Universal | 9.05±2.67[a] | 7.62-10.47 | 10.30±3.55[a] | 8.41-12.19 |
| | MDP | 10.20±2.51[a] | 8.86-11.53 | 10.10±3.58[a] | 8.19-12.01 |
| Cast | Universal* | 7.11±1.49[b] | 6.31-7.90 | 9.70±2.31[a] | 8.47-10.93 |
| | MDP | 7.40±3.26[b] | 5.67-9.14 | 8.95±1.96[a] | 7.90-9.99 |

Superscript letters (a,b,c) within the same column indicate Tukey HSD groupings, with means sharing a letter are not significantly different. An asterisk indicates a significant within-row difference (t-test with Holm correction). CI, confidence interval; SD, standard deviation; MDP, 10-methacryloyloxydecyl dihydrogen phosphate; MPa, megapascal.

frameworks. SLM Co-Cr often shows mechanical properties comparable to or superior to cast Co-Cr and has been validated for orthodontic components [26,27]. In contrast, Ni-Cr was chosen for conventional casting because it is the standard base-metal casting alloy for small precision parts, and its lower melting temperature than Co-Cr favors mold filling and limits investment reactions in thin sections [28,29]. Nonetheless, comparative studies report no significant castability

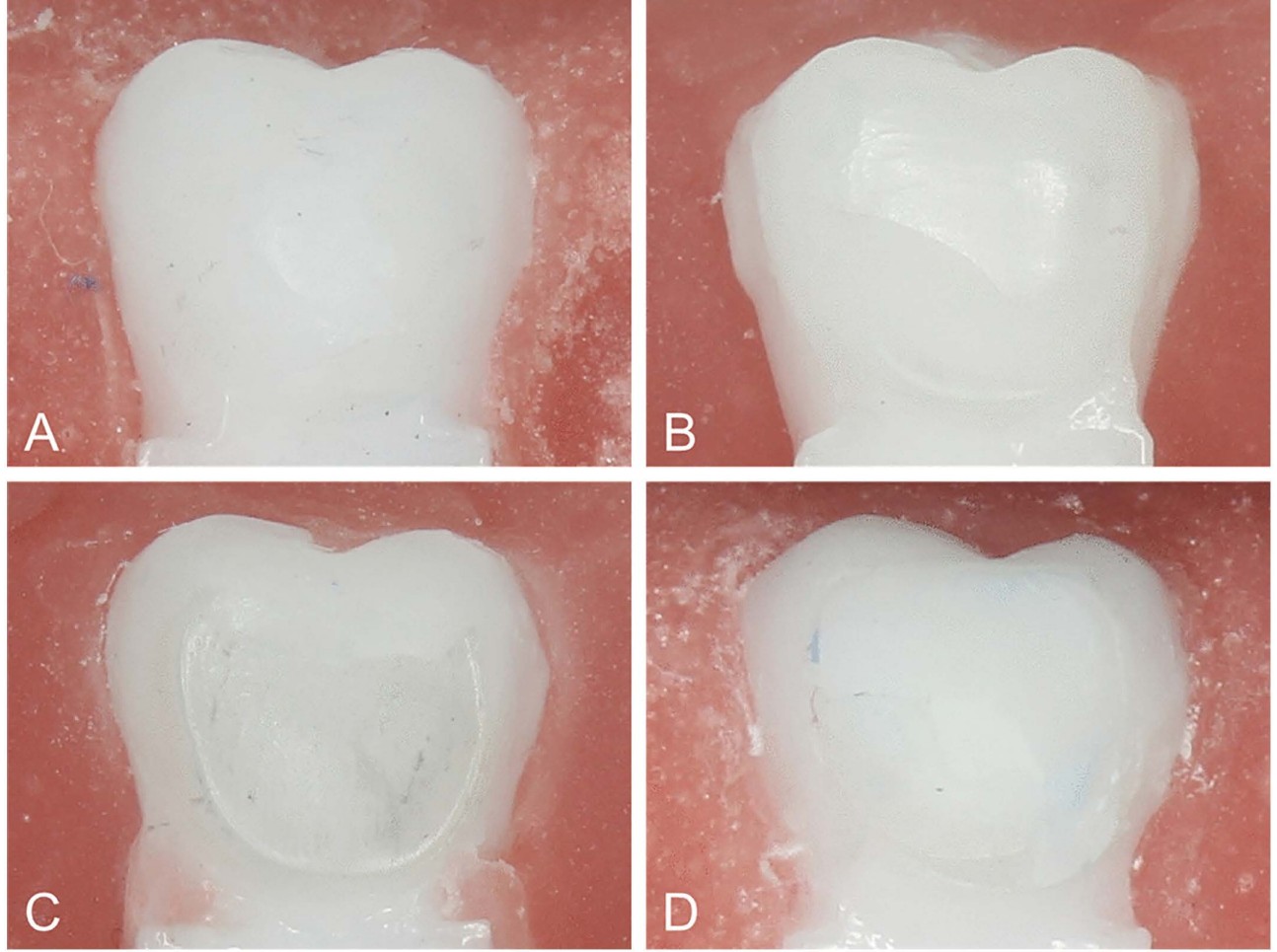

**Fig 4. Representative images of Adhesive Remnant Index (ARI) scores.** A, ARI 0: no adhesive remaining on the zirconia surface. B, ARI 1: less than 50% of the adhesive remaining on the zirconia surface. C, ARI 2: more than 50% of the adhesive remaining on the zirconia surface. D, ARI 3: all adhesive remaining on the zirconia surface.

difference between Ni-Cr and Co-Cr when casting parameters are optimized; our pairing, therefore, reflects common clinical-laboratory practice rather than an a priori expectation of superiority [28]. We selected 3Y-TZP zirconia as the bonding substrate because its relatively high opacity increases light attenuation, enabling a stringent comparison of light-cure versus dual-cure polymerization in the deeper adhesive fraction [5].

After airborne-particle abrasion, the lowest Ra on zirconia and higher Ra on metals align with the greater Vickers hardness (HV) of zirconia (1200–1400 HV) relative to typical Ni-Cr (200–430 HV) and Co-Cr (330–540 HV) [30–32]. Harder substrates develop shallower asperities under identical blasting, while additively manufactured Co-Cr tends to show particle-attached, layerwise irregularities that elevate as-built roughness versus cast alloys. In SEM images, the nearly dry appearance of the MDP primer after air-drying is consistent with its volatile and unfilled formulation. Because the specimens were coated under vacuum before imaging, residual solvents would have been further removed, potentially accentuating the dry appearance. Consequently, the dark primer remnants are mainly restricted to micro-pits, with more of the ceramic substrate visible. In contrast, the universal adhesive's water-containing, hydrophilic, and filler-reinforced matrix promotes surface wetting and yields the appearance of a more continuous post-drying film (uniform dark layer) on metallic

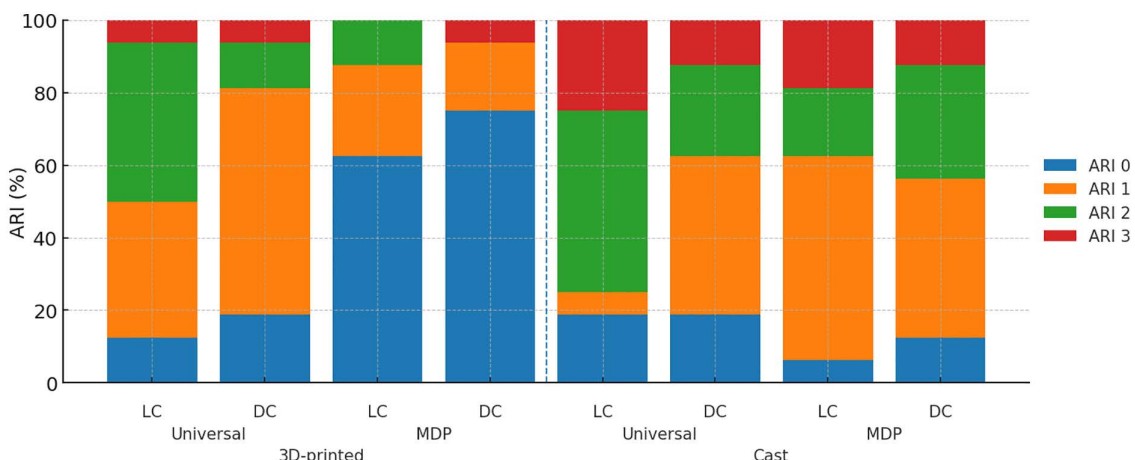

**Fig 5. Stacked bar chart showing the distribution of Adhesive Remnant Index (ARI) scores (0-3) by bracket fabrication modality, primer, and adhesive polymerization mode.**

substrates. On zirconia, however, limited silane reactivity and solvent-monomer phase behavior in one-bottle systems can promote de-wetting or phase separation during evaporation, producing discontinuous coverage with brighter exposed islands of the base ceramic [33,34].

For the main effect of fabrication method, 3D-printed brackets exhibited greater bond strength than cast brackets, plausibly due to their rougher post-abrasion surface. Consistently, prior work has reported higher bond strengths to Co–Cr than to Ni–Cr across multiple resin-cement systems [35]. The lack of a primer main effect can be explained by opposing mechanisms. A dedicated MDP primer, which is rich in phosphate monomer and with fewer non-reactive constituents, would be expected to outperform a multipurpose universal system on zirconia and metal oxides [36,37]. However, the universal adhesive's hydrophilic, water-containing, filler-reinforced matrix preserves surface wettability and film continuity after air-drying, offsetting any advantage of the higher MDP fraction [38,39]. Finally, the lack of a polymerization-mode effect is plausible given competing mechanisms. Light is attenuated by opaque metal and zirconia and may under-cure the deepest adhesive fraction, yet dual-cure systems chemically compensate in shadowed regions, whereas the photo-activated outer layer in light-cure groups can achieve high conversion, netting similar bond strength [40]. Future work should directly assess the degree of conversion beneath opaque substrates using Fourier-transform infrared spectroscopy (FTIR) to verify and generalize the present findings.

Printed-cast differences in bond strength appeared only under light-cure adhesive, with no between-fabrication-method contrasts under dual-cure cement. Under light cure, bond strength appears to be driven predominantly by micromechanical retention, therefore, the rougher post-abrasion surface of 3D-printed brackets promotes greater mechanical interlocking than cast. By contrast, dual-cure cements contribute more via chemical coupling (MDP within the cement) and self-curing in light-shadowed regions of metal and zirconia, reducing dependence on surface topography.

Across groups, the predominance of ARI 0–1 indicates failures closer to the zirconia-adhesive interface. This pattern is expected because abraded zirconia remains smoother than the metal bracket bases, whereas the latter's higher post-abrasion roughness yields stronger micromechanical interlocking and bracket-side retention. Furthermore, phosphate-based primers can chemisorb to native metal oxides more effectively than on zirconia, thereby reinforcing the bracket-adhesive junction [37,41]. Consistent with this mechanism, the 3D-printed groups with rougher bases exhibited lower ARI (more resin on brackets) together with higher bond strength, reflecting a failure locus shifted toward the zirconia-adhesive interface. Additionally, ARI decreased most for 3D-printed brackets with the MDP primer, whose

fast-evaporating and unfilled layer leaves sparse coverage on zirconia after air-drying. In contrast, the universal adhesive preserved surface wettability and a more continuous film, attenuating this effect.

In prior *in vitro* studies, bonding to zirconia is largely mechanochemically driven, with airborne-particle abrasion being essential [10]. Once an effective primer is applied, both light-cure orthodontic composites and dual-cure resin cements routinely achieve bond strengths of 8–15 MPa [12–14]. Without a primer, light-cure composites generally underperform and often fail after thermocycling, whereas dual-cure and self-adhesive systems containing functional phosphates can partially compensate [10,11,13]. Dedicated MDP primers and one-bottle universal adhesives tend to yield comparable shear bond strength on abraded zirconia. Reported differences are small and inconsistent across studies, with universal agents sometimes showing higher initial wettability but similar aged performance [12–14]. Direct head-to-head data on customized lingual brackets bonded to zirconia are scarce. On enamel, customized lingual brackets typically exhibit low-to-mid shear bond strength (2–6 MPa), yet their enlarged bases yield debonding forces that are equivalent to or can exceed those of stock brackets [15,19]. The lower bond strength in those reports likely reflects that an MDP-containing primer was not applied to the metal bracket base.

A clinically accepted threshold for bracket retention is 6–8 MPa [25,42]. In our model, most combinations clustered around 9–10 MPa, and even the lowest mean exceeded 7 MPa, satisfying this threshold. Because the customized lingual bases were larger than stock brackets, the same nominal shear stress translates into higher debond forces. Therefore, clinically acceptable performance can be achieved without targeting very high MPa values.

This study has several strengths. The 2 × 2 × 2 factorial design with equal allocation and an a priori power analysis enabled simultaneous estimation of main effects and interactions with balanced precision. Surface preparation and zirconia glazing were standardized, and complementary Ra measurements and SEM micrographs provided a mechanistic context to the bond-strength outcomes. Adhesive remnant evaluation was performed with blinded scoring, and the tested materials (MDP primer, universal adhesive, light-cure composite, dual-cure cement) reflect common clinical practice for bonding to glazed zirconia.

This study has limitations that warrant consideration. It was conducted in an *in vitro* setting with static shear loading after 2000 thermocycles and did not include cyclic fatigue, chewing simulation, and pH fluctuations. The thermocycling regimen should be interpreted as an early-aging proxy rather than a simulation of the entire duration of fixed-appliance therapy. A single zirconia and a single airborne-particle abrasion protocol were used, which may limit generalizability to other zirconia systems, glaze conditions, and surface treatments. The materials tested were restricted to one MDP primer, one universal adhesive, one light-cure orthodontic composite, and one dual-cure resin cement, so findings may not extend to other brands or newer chemistries. Potential variability in bracket base area and the lack of direct quantification of the effective bonded area could influence the calculated bond strength. Degree of conversion beneath opaque substrates, adhesive film thickness, and wettability/contact angle were not measured. Flat zirconia substrates were used rather than anatomic restorations, and tensile bond strength was not evaluated. Furthermore, the fabrication route and alloy were not fully orthogonal, so differences attributed to "fabrication" may partly reflect alloy-specific topographic or chemical effects. Future investigations addressing multiple alloys, substrates, surface protocols, and loading modes will be needed to confirm generalizability.

In practical terms, for bonding customized lingual brackets to glazed zirconia, manufacturing modality was the main driver of retention in this model, albeit with a borderline effect on bond strength. Additively manufactured Co–Cr bases showed higher shear bond strength than cast Ni–Cr, particularly with light-cure adhesives, with differences attenuated under dual-cure cements. Primer choice did not meaningfully change mean bond strength; thus, selection can be guided by handling and workflow. Notably, 3D-printed bases were associated with lower ARI levels than cast (especially with MDP primer), which may reduce chairside cleanup of the zirconia surface. Routine airborne-particle abrasion remains essential for zirconia, and curing strategies should account for light attenuation under opaque ceramics.

## Conclusions

Within an *in vitro* 2×2×2 design on glazed, sandblasted zirconia, shear bond strength showed a borderline main effect of fabrication, with additively manufactured Co–Cr customized bases associated with higher values than cast Ni–Cr counterparts. Primer type and polymerization mode did not show meaningful main effects, and no interactions were detected. Failures occurred predominantly at or near the zirconia–adhesive interface. Given the modest effect size and *in vitro* context, clinical implications remain uncertain; at most, the base manufacturing method may be considered when other factors are comparable, whereas primer and curing mode can be selected based on handling and workflow until confirmed in clinically representative studies.

## Supporting information

**S1 File. Dataset.**
(XLSX)

## Author contributions

**Conceptualization:** Viet Anh Nguyen.

**Data curation:** Ngo The Minh Pham, Minh Ngoc Tran, Thi Bich Ngoc Ha.

**Formal analysis:** Ngo The Minh Pham, Minh Ngoc Tran, Thi Bich Ngoc Ha.

**Funding acquisition:** Viet Anh Nguyen.

**Investigation:** Ngo The Minh Pham, Minh Ngoc Tran, Thi Bich Ngoc Ha, Thi Quynh Trang Vuong.

**Methodology:** Viet Anh Nguyen, Thi Quynh Trang Vuong.

**Project administration:** Viet Anh Nguyen.

**Resources:** Viet Anh Nguyen.

**Software:** Viet Anh Nguyen.

**Supervision:** Viet Anh Nguyen.

**Validation:** Viet Anh Nguyen.

**Visualization:** Viet Anh Nguyen.

**Writing – original draft:** Viet Anh Nguyen.

**Writing – review & editing:** Viet Anh Nguyen.

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
