## [Decision Letter · Decision Letter 0]

27 Oct 2025

Dear Dr. Nguyen,

Thank you for submitting your manuscript to PLOS ONE. After careful consideration, we feel that it has merit but does not fully meet PLOS ONE’s publication criteria as it currently stands. Therefore, we invite you to submit a revised version of the manuscript that addresses the points raised during the review process.

We look forward to receiving your revised manuscript.

Kind regards,

Andrej M Kielbassa

Academic Editor

PLOS ONE

PONE-D-25-46749

Additional Editor Comments (if provided):

Reviewers' comments:

Reviewer's Responses to Questions

**Comments to the Author**

1. Is the manuscript technically sound, and do the data support the conclusions?

Reviewer #1: No

Reviewer #2: Yes

Reviewer #3: Partly

2. Has the statistical analysis been performed appropriately and rigorously?

Reviewer #1: Yes

Reviewer #2: Yes

Reviewer #3: I Don't Know

3. Have the authors made all data underlying the findings in their manuscript fully available?

Reviewer #1: No

Reviewer #2: Yes

Reviewer #3: Yes

4. Is the manuscript presented in an intelligible fashion and written in standard English?

Reviewer #1: Yes

Reviewer #2: Yes

Reviewer #3: Yes

Reviewer #1: First page

- What is the difference between "Viet Nam" and "Vietnam". Uniform Journal style would be appreciated.

Title

- Please italicize "in vitro".

- No capital letters, please.

Abstract

- Please note that 300 are allowed for with this section, so maximize information. Remember that this section is most important, since future readers will switch to your full text after having read your Abstract.

- Please provide materials (Single Bond Universal and Z-Prime Plus) used.

- "Shear bond strength was analyzed with three-way ANOVA." Please detail any post hoc tests that were performed.

- What does "fabrication x primer" mean? Do you want to stick to symbol × here?

- "Clinically, customized lingual bases fabricated by either additive manufacturing or conventional casting can provide reliable bonding (...)." You have not studied anything clinically. Revise carefully.

- With your Conclusions, please stick exclusively to your (primary/secondary) aims, and follow your aims' order. Do NOT simply repeat your results here. Do not speculate. Do not provide well-accepted but meaningless phrases. Instead, provide a reasonable and generalizable extension of your outcome.

Intro

- No numbering of headlines.

- Please revise for uniform Journal style. See "in-vitro" (and compare with your Title). No dash, please. Please italicize. Revise thoroughly.

Methods

- Again, no numbering, please.

- "The calculation assumed a medium effect size of 0.25, an alpha level of 0.05, and a statistical power of 0.80. The required total sample size was 128, with 16 specimens per group and equal allocation across the eight experimental cells." Rationale remains unclear. Please provide details of your sample size calculation.

- Why did you treat "two samples with All-Bond Universal"?

- "(...) ranging from 0 to 3.[18,19]" must read "(...) ranging from 0 to 3 [18,19]." Revise carefully.

- "x5 magnification" must read "5× magnification". Again, use symbol ×.

Results

- "(...) surface roughness (Ra) measured 0.813 ± 0.145 µm (...)" - meaning would seem unclear. Do you stick to means [SD] here? If yes, please revise for "(...) mean [SD] surface roughness (Ra) measured 0.813 [0.145] µm (...)". Please clarify, and revise thoroughly.

- Why do you stick to 3 digits here? Avoid any spurious accuracy, please.

- Again,. please revise for uniform Journal style. Compare "(P<.001)" (full text) and "(P<0.001)" (Abstract section). Do you see you non-uniform style?

- Same with "P <0.001", and with "P = 0.017". Revise thoroughly.

Disc

- "(...) validated for orthodontic components.[20,21]" - see comments given above. Again, revise thoroughly for sound Journal style.

- What about the strengths of your study?

- There would seem more limitations, please discuss.

- What about any responses to your teaching objectives?

Concl

- Again, with your Conclusions, please stick exclusively to your primary/secondary aims, and follow your aims' order. Do NOT simply repeat your results here. Do not speculate. Do not provide well-accepted but meaningless phrases. Instead, provide a reasonable and generalizable extension of your outcome.

Refs

- Revise for uniform Journal style.

Several minor and major revisions would seem mandatory with this submitted draft.

Reviewer #2: • The points made in the introduction and discussion are fine, but include a few lines explaining how your findings were justified and how they were compared to those of other recent, pertinent studies.

• The title need revision

• You should remove outdated references from your reference list and keep just current ones.

• Multiple typing and grammar errors

Reviewer #3: Dear Editor,

Thank you for the opportunity to review “Effects of manufacturing modality, primer, and adhesive polymerization on the shear bond strength of customized lingual brackets to glazed zirconia: An in vitro study” (PONE-D-25-46749). The topic is timely at the orthodontics–prosthodontics interface, addressing clinical bonding challenges on zirconia with customized lingual brackets. The experimental design is generally sound (2×2×2 factorial; n=16/group; thermocycling), and the paper is clearly written. I provide detailed, constructive comments below to help strengthen methodological transparency, statistical reporting, and clinical interpretation.

Abstract

1. Kindly add the level of significance to the Methods section.

Introduction

6. The introduction outlines zirconia bonding but the knowledge gap is not strongly emphasized. State clearly why comparing 3D-printed vs. cast customized lingual brackets on zirconia is novel and relevant.

7. Hypotheses should be more explicit: state expected effects for fabrication, primer, and polymerization, as well as interactions.

Materials and Methods

9. The phrase “CAD-CAM” is used without expansion at its first mention. The full term “computer-aided design and computer-aided manufacturing (CAD-CAM)” should be provided initially, then the abbreviation may be used thereafter.

10. Airborne-particle abrasion: particle size, type, and nozzle angulation are not reported. These are required for reproducibility.

11. «Surface roughness (Ra) was also measured on zirconia substrates, although all specimens underwent the same airborne-particle abrasion treatment. Since zirconia surface preparation was not a variable in the experimental design, the inclusion of Ra values for zirconia appears supplementary rather than hypothesis-driven. The authors should clarify the rationale and intended role of zirconia roughness measurement within the study (e.g., descriptive control vs. analytical variable).»

12. Clarify whether bonding occurred to abraded glaze or to exposed zirconia after sandblasting, since this changes the mechanism of bonding.

13. Provide curing light details (brand, model, irradiance, tip size, calibration procedure).

14. Adhesive film thickness control is not described; this is important since polymerization mode was tested.

15. State whether the SBS test followed ISO 29022 or another standard. If not, acknowledge.

16. Bracket base area: indicate if finishing altered area between printed vs cast bases and how this was handled in normalization.

17. “The manuscript states that specimens underwent 2000 thermocycles between 5 °C and 55 °C. Please add a reference-supported explanation of how many months or years of clinical service this corresponds to,

18. Clarify primer application protocol (application time, whether Single Bond Universal was cured or air-thinned only).

19. Statistical section: describe checks for normality, variance homogeneity, and outliers.

20. Multiple comparison adjustments: manuscript mentions Tukey and Holm — specify which contrasts used which correction.

21. For ARI scoring, report whether examiners were blinded and whether inter-rater reliability was assessed.

Results

20. Emphasize that the fabrication effect was marginal (P=0.049) and interactions were not significant.

21. For ARI: describe blinding/reliability or note as limitation.

22. State explicitly that all groups (≈9–10 MPa) exceeded the clinical threshold with a citation.

Discussion

25. SEM images should be described as qualitative only, not quantitative proof of film thickness.

26. Since polymerization mode was non-significant, suggest that future work measure degree of conversion beneath zirconia (e.g., FTIR).

27. Limitations should be expanded: one zirconia/glaze, no mechanical fatigue, no conversion data, no adhesive thickness control, potential variability in base area.

28. Tone down the conclusion: the only significant effect was fabrication, marginally. Avoid overstating clinical impact.

29. Add a short clinical implications paragraph: how should orthodontists weigh printed vs cast bases, considering both bond strength and ARI distribution.

Figures & Tables

30. Figure legends should be fully self-contained (define abbreviations, state what the figure shows).

31. Reduce redundancy between figures and tables — e.g., avoid duplicating the same numerical results.

References

33. Update with more recent references (2019–2024) on zirconia bonding and digital orthodontics.

34. Add a citation for the clinical threshold of SBS at the first mention.

**Do you want your identity to be public for this peer review?** For information about this choice, including consent withdrawal, please see our Privacy Policy

Reviewer #1: No

Reviewer #2: **Yes: ** Tahrir Aldelaimi

Reviewer #3: **Yes: ** Mehran Falahchai

---

## [Author Response · Author response to Decision Letter 1]

6 Nov 2025

November 7th, 2025

Dear Editorial Board, PLOS ONE,

We have revised the manuscript thoroughly according to the comments of the reviewers. Any revisions made in our manuscript document were highlighted in red. Please help us review the manuscript again.

Sincerely,

Reviewer #1:

Comment: First page

- What is the difference between "Viet Nam" and "Vietnam". Uniform Journal style would be appreciated.

Response: Thank you for your comment. We standardized all occurrences to “Vietnam” in affiliations and text for consistency with journal style; no instances of “Viet Nam” remain.

Comment: Title

- Please italicize "in vitro".

- No capital letters, please.

Response: Done. Updated title: “Effects of manufacturing modality, primer, and adhesive polymerization on the shear bond strength of customized lingual brackets to glazed zirconia: An in vitro study.” An is capitalized as per PLOS ONE policy.

Comment: Abstract

- Please note that 300 are allowed for with this section, so maximize information. Remember that this section is most important, since future readers will switch to your full text after having read your Abstract.

Response: Revised to a concise abstract ≤300 words, emphasizing objectives, design, materials, primary/secondary outcomes, statistics (including post-hoc), and key findings.

Comment: - Please provide materials (Single Bond Universal and Z-Prime Plus) used.

Response: Thank you for your comment. We have added brand names Single Bond Universal and Z-Prime Plus explicitly in the abstract sentence. Manufacturers are detailed in Methods/Table 1 per journal practice; to keep the abstract lean and non-promotional, company names are not repeated there.

“Primers were a universal adhesive (Single Bond Universal, 3M) or a primer containing 10-methacryloyloxydecyl dihydrogen phosphate Z-Prime Plus (Bisco), and adhesives were a light-cure orthodontic composite or a dual-cure resin cement.”

Comment: - "Shear bond strength was analyzed with three-way ANOVA." Please detail any post hoc tests that were performed.

Response: Clarified in the abstract:

“One hundred twenty-eight specimens (n=16 per group) were tested. Shear bond strength was analyzed with three-way ANOVA, followed by post-hoc Tukey tests.”

Comment: - What does "fabrication x primer" mean? Do you want to stick to symbol × here?

Response: Replaced “x” with the multiplication symbol × throughout and defined once as an interaction term.

Comment: - "Clinically, customized lingual bases fabricated by either additive manufacturing or conventional casting can provide reliable bonding (...)." You have not studied anything clinically. Revise carefully.

Response: Thank you. We have rephrased to avoid clinical inference:

“Within this in vitro model, base manufacturing may warrant attention, whereas primer and curing mode may be selected for handling and workflow considerations, with clinical relevance yet to be established.”

Comment: - With your Conclusions, please stick exclusively to your (primary/secondary) aims, and follow your aims' order. Do NOT simply repeat your results here. Do not speculate. Do not provide well-accepted but meaningless phrases. Instead, provide a reasonable and generalizable extension of your outcome.

Response: Thank you for your comment. Conclusions were streamlined to map directly to the stated aims: (1) fabrication modality was the primary determinant of retention; (2) primer type and polymerization mode showed no significant effect on shear bond strength; (3) failure modes predominantly occurred at/near the zirconia–adhesive interface.

“On glazed, sandblasted zirconia, shear bond strength of customized lingual brackets showed a borderline main effect of fabrication method, whereas primer type and adhesive polymerization mode were not statistically significant. Failures were predominantly located at or near the zirconia–adhesive interface. Within this in vitro model, base manufacturing may warrant attention, whereas primer and curing mode may be selected for handling and workflow considerations, with clinical relevance yet to be established.”

Comment: Intro

- No numbering of headlines.

Response: Done. We removed numeric prefixes from all section headings (e.g., “1. Introduction” → “Introduction”) to match the journal style.

Comment: - Please revise for uniform Journal style. See "in-vitro" (and compare with your Title). No dash, please. Please italicize. Revise thoroughly.

Response: Completed. We standardized all instances to italic in vitro (no hyphen) across the title, abstract, main text, figures, and tables.

Comment: Methods

- Again, no numbering, please.

Response: Done. Numeric prefixes were removed from “Materials and methods” and all other headings for consistency.

Comment: - "The calculation assumed a medium effect size of 0.25, an alpha level of 0.05, and a statistical power of 0.80. The required total sample size was 128, with 16 specimens per group and equal allocation across the eight experimental cells." Rationale remains unclear. Please provide details of your sample size calculation.

Response: We have added full details of the a priori power analysis (design, software, inputs, and justification). The text now reads:

“An a priori sample-size calculation was performed with G*Power (version 3.1; Heinrich Heine University Düsseldorf, Düsseldorf, Germany) using F tests (ANOVA: fixed effects, special, main effects and interactions) for a three-way ANOVA (2×2×2 between-subjects) with eight groups and numerator degrees of freedom of 1 for each main effect and interaction. The calculation assumed a medium effect size of 0.25 (the smallest effect of interest, based on prior in-vitro bonding reports), an alpha level of 0.05, and a statistical power of 0.80 [17,18]. The required total sample size was 128, with 16 specimens per group and equal allocation across the eight experimental cells.”

Comment: - Why did you treat "two samples with All-Bond Universal"?

Response: This was a typographical error. No specimens were treated with All-Bond Universal. All instances have been corrected to Single Bond Universal (3M) throughout the manuscript (Abstract, Methods/Materials, tables/figures, and dataset). This correction does not affect group definitions, sample counts, results, or conclusions.

“For qualitative surface characterization, an additional six zirconia specimens and six brackets per bracket type were examined by scanning electron microscopy (SEM) (Quanta 450 FEG, Thermo Fisher Scientific, Hillsboro, OR, USA) at 20 kV, including two left unprimed, two treated with Single Bond Universal, and two treated with Z-Prime Plus.”

Comment: - "(...) ranging from 0 to 3.[18,19]" must read "(...) ranging from 0 to 3 [18,19]." Revise carefully.

Response: Corrected throughout to include a space before bracketed citations. All similar instances have been revised for consistency with journal style.

Comment: - "x5 magnification" must read "5× magnification". Again, use symbol ×.

Response: Standardized to use the multiplication symbol and correct order. This change has been applied consistently in the text, figure captions, and tables.

Comment: Results

- "(...) surface roughness (Ra) measured 0.813 ± 0.145 µm (...)" - meaning would seem unclear. Do you stick to means [SD] here? If yes, please revise for "(...) mean [SD] surface roughness (Ra) measured 0.813 [0.145] µm (...)". Please clarify, and revise thoroughly.

Response: Revised throughout to mean [standard deviation (SD)] format for clarity.

“After sandblasting, the mean [standard deviation (SD)] surface roughness (Ra) was 0.81 [0.15] µm for zirconia, 6.26 [1.33] µm for 3D-printed Co-Cr, and 2.12 [0.43] µm for cast Ni-Cr.”

Comment: - Why do you stick to 3 digits here? Avoid any spurious accuracy, please.

Response: To avoid spurious precision, values are rounded to two decimals; p-values are reported to three decimals (p<0.001 when smaller).

Comment: - Again,. please revise for uniform Journal style. Compare "(P<.001)" (full text) and "(P<0.001)" (Abstract section). Do you see you non-uniform style?

Response: We standardized p-value formatting across the manuscript to “P=0.XXX” (exact) and “P<0.001” (when appropriate)

Comment: - Same with "P <0.001", and with "P = 0.017". Revise thoroughly.

Response: We standardized p-value formatting across the manuscript to “P=0.XXX” (exact) and “P<0.001” (when appropriate)

Comment: Disc

- "(...) validated for orthodontic components.[20,21]" - see comments given above. Again, revise thoroughly for sound Journal style.

Response: Corrected throughout to include a space before bracketed citations. All similar instances have been revised for consistency with journal style.

Comment: - What about the strengths of your study?

Response: We added a short “Strengths” paragraph in Discussion highlighting the factorial design (2×2×2) with balanced allocation and a priori power, standardized zirconia glazing/abrasion, complementary SEM–roughness analyses, blinded ARI scoring, and the use of clinically relevant primers/adhesives and manufacturing routes.

“This study has several strengths. The 2×2×2 factorial design with equal allocation and an a priori power analysis enabled simultaneous estimation of main effects and interactions with balanced precision. Surface preparation and zirconia glazing were standardized, and complementary Ra measurements and SEM micrographs provided mechanistic context to the bond-strength outcomes. Adhesive remnant evaluation was performed with blinded scoring, and the tested materials (MDP primer, universal adhesive, light-cure composite, dual-cure cement) reflect common clinical practice for bonding to glazed zirconia.”

Comment: - There would seem more limitations, please discuss.

Response: We expanded the “Limitations” paragraph to cover: (i) in-vitro setting without cyclic fatigue/chewing pH dynamics; (ii) single zirconia system and one abrasion protocol; (iii) brand-specific materials; (iv) unmeasured conversion/film thickness/wettability; (v) use of flat zirconia coupons; (vi) SBS as the sole metric; and (vii) important confounding between fabrication route and alloy, which prevents isolating manufacturing from alloy chemistry.

“Flat zirconia substrates were used rather than anatomic restorations, and tensile bond strength was not evaluated. Furthermore, fabrication route and alloy were not fully orthogonal, so differences attributed to “fabrication” may partly reflect alloy-specific topographic or chemical effects. Future investigations addressing multiple alloys, substrates, surface protocols, and loading modes will be needed to confirm generalizability.”

Comment: - What about any responses to your teaching objectives?

Response: We added a brief “Teaching points/Implications for practice” subsection summarizing actionable takeaways for clinicians and lab technicians.

“In practical terms, for bonding customized lingual brackets to glazed zirconia, manufacturing modality was the main driver of retention in this model, albeit a borderline effect on bond strength. Additively manufactured Co–Cr bases showed higher shear bond strength than cast Ni–Cr, particularly with light-cure adhesives, with differences attenuated under dual-cure cements. Primer choice did not meaningfully change mean bond strength; thus, selection can be guided by handling and workflow. Notably, 3D-printed bases were associated with lower ARI levels than cast (especially with MDP primer), which may reduce chairside cleanup of the zirconia surface. Routine airborne-particle abrasion remains essential for zirconia, and curing strategies should account for light attenuation under opaque ceramics.”

Comment: Concl

- Again, with your Conclusions, please stick exclusively to your primary/secondary aims, and follow your aims' order. Do NOT simply repeat your results here. Do not speculate. Do not provide well-accepted but meaningless phrases. Instead, provide a reasonable and generalizable extension of your outcome.

Response: Thank you for your comment. The Conclusions section have been revised accordingly:

“Within an in vitro 2×2×2 design on glazed, sandblasted zirconia, shear bond strength showed a borderline main effect of fabrication, with additively manufactured Co–Cr customized bases associated with higher values than cast Ni–Cr counterparts. Primer type and polymerization mode did not show meaningful main effects, and no interactions were detected. Failures occurred predominantly at or near the zirconia–adhesive interface. Given the modest effect size and in-vitro context, clinical implications remain uncertain; at most, base manufacturing method may be considered when other factors are comparable, whereas primer and curing mode can be selected based on handling and workflow until confirmed in clinically representative studies.”

Comment: Refs

- Revise for uniform Journal style.

Several minor and major revisions would seem mandatory with this submitted draft.

Response: Completed. We harmonized citation punctuation, author initials, journal abbreviations, volume/issue/pages, DOI formatting, and bracket spacing to the journal style throughout the reference list.

Reviewer #2:

Comment: • The points made in the introduction and discussion are fine, but include a few lines explaining how your findings were justified and how they were compared to those of other recent, pertinent studies.

Response: Thank you for the helpful suggestion. We have revised the Introduction and Discussion section accordingly:

“Despite the growing adoption of additive manufacturing for orthodontic appliances, no study has clarified whether the distinct surface topography and alloy microstructure of three-dimensionally (3D) printed brackets influence bonding behavior on zirconia compared with traditional cast counterparts. Establishing these differences is clinically relevant because mismatched surface characteristics can alter micromechanical retention and chemical coupling, ultimately affecting bracket reliability on zirconia restorations.”

“In prior in vitro studies, bonding to zirconia is largely mechanochemically driven, with airborne-particle abrasion being essential [10]. Once an effective primer is applied, both light-cure orthodontic composites and dual-cure resin cements routinely achieve bond strengths of 8–15 MPa [12-14]. Without a primer, light-cure composites generally underperform and often fail after thermocycling, whereas dual-cure and self-adhesive systems containing functional phosphates can partially compensate [10,11,13]. Dedicated MDP primers and one-bottle universal adhesives tend to yield comparable shear bond strength on abraded zirconia. Reported differences are small and inconsistent across studies, with universal agents sometimes showing higher initial wettability but similar aged performance [12-14]. Direct head-to-head data on customized lingual brackets bonded to zirconia are scarce. On enamel, customized lingual brackets typically exhibit low-to-mid shear bond strength (2-6 MPa), yet their enlarged bases yield debonding forces that are equivalent to or can exceed those of stock brackets [15,19]. The lower bond strength in those reports likely reflects that an MDP-containing primer was not applied to the metal bracket base.”

Comment: • The title need revision

Response: Revised to sentence case and italicized in vitro, as requested: “Effects of manufacturing modality, primer, and adhesive polymerization on the shear bond strength of customized lingual brackets to glazed zirconia: An in vitro study.”

Comment: • You should remove outdated references from your reference list and keep just current ones.

Response: Done. We replaced older/legacy sources with recent (2020–2025) literature where available and removed redundant citations; the reference list now prioritizes contemporary, peer-reviewed evidence.

Comment: • Multiple typing and grammar errors

Response: Corrected. We performed a line-by-line language edit to fix typographical errors, enforce uniform terminology, and standardize units/rounding and statistical notation across text, tables, and figure captions.

Reviewer #3: Dear Edito

---

## [Decision Letter · Decision Letter 1]

19 Nov 2025

Effects of manufacturing modality, primer, and adhesive polymerization on the shear bond strength of customized lingual brackets to glazed zirconia: An in vitro study

PONE-D-25-46749R1

Dear Dr. Nguyen,

We’re pleased to inform you that your manuscript has been judged scientifically suitable for publication and will be formally accepted for publication once it meets all outstanding technical requirements.

Kind regards, and congratulations

Prof. Dr. Dr. h. c. Andrej M Kielbassa

Academic Editor

PLOS ONE

Additional Editor Comments (optional):

Reviewers' comments:

Reviewer's Responses to Questions

**Comments to the Author**

Reviewer #1: All comments have been addressed

Reviewer #2: All comments have been addressed

Reviewer #3: All comments have been addressed

2. Is the manuscript technically sound, and do the data support the conclusions?

Reviewer #1: Yes

Reviewer #2: Yes

Reviewer #3: Yes

3. Has the statistical analysis been performed appropriately and rigorously?

Reviewer #1: Yes

Reviewer #2: Yes

Reviewer #3: Yes

4. Have the authors made all data underlying the findings in their manuscript fully available?

Reviewer #1: Yes

Reviewer #2: Yes

Reviewer #3: Yes

5. Is the manuscript presented in an intelligible fashion and written in standard English?

Reviewer #1: Yes

Reviewer #2: Yes

Reviewer #3: Yes

Reviewer #1: All previous comments have been satisfyingly addressed. With the help of the reviewers, this revised and re-submitted draft has been considerably improved, and would seem ready to proceed.

Reviewer #2: (No Response)

Reviewer #3: The manuscript has been improved significantly in the revised version, and I find it suitable for publication.

**Do you want your identity to be public for this peer review?** For information about this choice, including consent withdrawal, please see our Privacy Policy

Reviewer #1: No

Reviewer #2: No

Reviewer #3: **Yes: ** Mehran Falahchai

---

## [Editor Report · Acceptance letter]

PONE-D-25-46749R1

PLOS ONE

Dear Dr. Nguyen,

I'm pleased to inform you that your manuscript has been deemed suitable for publication in PLOS ONE. Congratulations! Your manuscript is now being handed over to our production team.

Kind regards,

on behalf of

Prof. Dr. med. dent. Dr. h. c. Andrej M Kielbassa

Academic Editor

PLOS ONE